# Pentavalent Antimony Associated with G-CSF in the Treatment of Cutaneous Leishmaniasis Caused by *Leishmania (Viannia) braziliensis*


**DOI:** 10.3390/pathogens13040301

**Published:** 2024-04-04

**Authors:** Carvel Suprien, Luiz H. Guimarães, Lucas P. de Carvalho, Paulo R. L. Machado

**Affiliations:** 1Postgraduate Program in Health Sciences, Faculty of Medicine, Federal University of Bahia, Salvador 40026-010, Bahia, Brazil; supriencarvel1@gmail.com (C.S.); lucas.carvalho@bahia.fiocruz.br (L.P.d.C.); 2National Institutes of Science and Technology in Tropical Diseases, Ministry of Science and Technology, Salvador, Bahia, Brazil; luizhenriquesg@yahoo.com.br; 3Medicine School, Federal University of Recôncavo Bahia, Santo Antônio de Jesus 44380-000, Bahia, Brazil; 4Immunology Service of the Professor Edgard Santos University Hospital, Federal University of Bahia, Salvador 40110-060, Bahia, Brazil; 5Gonçalo Moniz Institute, Fiocruz, Salvador 40296-710, Bahia, Brazil

**Keywords:** Cutaneous leishmaniasis, treatment, granulocyte colony stimulating factor, meglumine antimoniate, *Leishmania Viannia braziliensis*

## Abstract

Cutaneous leishmaniasis (CL), caused by *Leishmania braziliensis,* in recent decades has shown decreasing cure rates after treatment with meglumine antimoniate (MA). Granulocyte colony-stimulating factor (G-CSF) is a cytokine associated with epithelialization and healing processes. Methods: This study compares the effectiveness of G-CSF associated with MA in the treatment of CL. A total of 32 patients aged between 18 and 50 years with CL confirmed for *L. braziliensis* were included in this study. G-CSF or placebo (0.9% saline) was applied by intralesional infiltration at four equidistant points on the edges of the largest ulcer on days 0 and 15 of treatment associated with intravenous MA. Results: Males predominated in the G-CSF group (59%), while females predominated in the control group (53%). Injuries to the lower limbs predominated in both study groups. The cure rate in the G-CSF group was 65% and in the control group it was 47%, 90 days after initiation of therapy. Conclusions: Our data indicate that the association of G-CSF with MA is not superior to MA monotherapy. Although not significant, the potential benefit of this combination deserves further investigation. The use of higher doses or other routes of application of G-CSF in a greater number of patients should contribute to a definitive response.

## 1. Introduction

Leishmaniasis is among the six most important infectious–parasitic diseases in the world and is considered neglected by the World Health Organization. Cutaneous leishmaniasis (CL) is the most common form of presentation of American tegumentary leishmaniasis (ATL), accounting for more than 90% of transmission cases in the endemic region of Corte de Pedra, Bahia [1]. The standard treatment of CL is with meglumine antimoniate (MA) at a dose of 15–20 mg/kg per day for 20 days, as recommended by the Brazilian Ministry of Health (MS) [2]. However, low cure rates have been described in patients [3], and a long period of 60 to 90 days is required for the healing of the ulcerative lesion, this indicates the need to use alternative drugs. Currently, alternatives include other parenteral drugs such as pentamidine and amphotericin B [4], whose uses are limited by toxicity or by the parenteral route, which hinders adherence and regularity of treatment in rural areas. Amphotericin B, pentamidine, and miltefosine can be used as second-choice drugs, but they are also toxic. Furthermore, the effectiveness of the treatment depends on the species of Leishmania involved in the infection, as some species are more resistant to certain drugs [5]. In this context, it is important to develop more effective treatments to increase the cure rate and reduce the morbidity and absenteeism caused by the disease.

The use of multidrug therapy in diseases caused by intracellular agents has been indicated for a long time to treat tuberculosis and leprosy [6,7]. Also, with respect to ATL there are several examples of the use of more than one drug. In CL and mucosal leishmaniasis (ML), tissue damage is related to the host’s immune response to the toxic action of the parasite. As there is evidence that the activation of T-cells and the frequency of T cells expressing TNF or IFN are associated with the size of the leishmaniasis ulcer [8], association with immunomodulatory agents has been used as coadjuvants in the treatment of CL and ML. In ML, the association of AM with pentoxifylline, a TNF inhibitor medication, is more effective than AM, reduces healing time and cures patients’ refractory to AM. [9,10]. GM-CSF has the property of modulating the immune response when associated with MA, is both subcutaneously and topically more effective than MA, and reduces healing time in CL [11,12]. However, GM-CSF was discontinued for several years and was replaced by granulocyte colony stimulating factor (G-CSF). G-CSF is a 19 kDa glycoprotein that stimulates the production of granulocytes by the bone marrow, stimulating proliferation, differentiation, and neutrophil function [13]. In addition, G-CSF also plays an important role in skin healing by activating keratinocyte proliferation and is produced by fibroblasts when interacting with these cells [14]. It has been used experimentally in patients with toxic epidermal necrolysis [15] and dystrophic epidermolysis bullosa [16] to accelerate the process of epithelialization and healing. Finally, G-CSF has anti-Th1 action and induces IL-10-producing regulatory cells in addition to negatively interfering with the function of CD8^+^ cytotoxic cells, which are recognized as important agents in the tissue and pathogenesis of CL [17,18]. All these G-CSF actions may be important in controlling the intense inflammation that implies the appearance and maintenance of ulcers as well as the stimulation of cutaneous healing. The objective of this study was to evaluate the efficacy and safety of the association between MA and G-CSF compared to MA plus placebo in the treatment of CL; we also aimed to verify whether the intralesional use of G-CSF modified systemic production of the cytokines that are associated with inflammatory activity.

## 2. Materials and Methods

This trial compared the efficacy of intravenous MA (Glucantime™; Sanofi Aventis) associated with intralesional rHu G-CSF (Filgrastine™; Blau) in the treatment of CL. Thirty-two patients aged between 18 and 50 years from the endemic region of Corte de Pedra - Bahia - Brazil were included. 

### 2.1. Endemic Area and Case Definition of CL

Patients were recruited at the Corte de Pedra Health Center, in Bahia, Northeast Brazil, an endemic area for *L. braziliensis* infection. CL was diagnosed by the presence of 1 or more ulcerative lesion(s) on the skin, with laboratory confirmation performed by detection of *L. braziliensis* DNA by polymerase chain reaction (PCR) or by histopathology showing amastigotes in the tissue. Women of childbearing age were included only after a negative beta human chorionic gonadotropin (HCG) test to exclude pregnancy and used parenteral contraceptives for 3 months.

### 2.2. Patient Selection

The inclusion criteria were men and women between the ages of 18 and 50 years who had 1 to 3 ulcers, a lesion between 20 and 50 mm in size in a single dimension and a period of between 30 and 90 days from the onset of the skin lesion. We did not include patients with previous CL treatment; patients with evidence of ML or DL; patients with severe kidney or heart disease; or patients with systemic infectious disease.

### 2.3. Sample Size, Randomization and Group Assignment

The total sample size of 32 patients was obtained considering a 30% variation in the cure rate in the control group compared to the intervention group, with an alpha of 0.05 and a power of 85% in the study group. Randomization was performed according to a computer list obtained from www.randomization.com and allocated patients into 2 groups: MA (Glucantime™; Sanofi Aventis) associated with placebo (control) and the other group was MA associated with intralesional rHu G-CSF (Filgrastine™; Blau). Two blinded physicians from the assigned group performed the physical examination and determined the therapeutic outcome. Patients and doctors were advised not to exchange information about treatment.

### 2.4. Histopathology, PCR and Leishmania Skin Test

All patients underwent biopsies from the edge of the ulcer, and 2 skin fragments were obtained for histopathological analysis and PCR. DNA isolation, purification and amplification were performed as described elsewhere [19]. Detection of the subgenus Viannia applied the primers GGGGTTGGTGTAATATAGTGG and CTAATTGTGCACG. The Leishmania-specific band consists of 120 base pairs and that for Viannia consists of 750 base pairs [20]. Leishmania skin test (LST): an intradermal injection of 0.1 mL of distilled water with 25 mg of antigen obtained from the *Leishmania amazonensis* strain (MHOM-BR-86BA-125) was administered in the left forearm. After 48 h, the largest diameter of induration was measured; LST was considered positive for induration greater than 5 mm.

### 2.5. Drug Administration

rHu G-CSF (Filgrastine™; Blau) (300 µg/mL) or placebo (0.9% saline) was applied by intralesional infiltration of 0.1 mL in 4 equidistant points on the edges of the largest ulcer on day 0 (D0—initiation of treatment) and 15 days after initiation of treatment (D15). All patients received standard systemic treatment with MA (Glucantime™; Sanofi Aventis); 20 mg/kg/day intravenously for 20 days.

### 2.6. Study Procedures

A complete blood count, aminotransferases (aspartate aminotransferase (AST) and alanine aminotransferase (ALT)), urea, creatinine and blood sugar levels were determined at D0, D15 and D60. Immunological studies were also carried out on days 0 and 15 (during treatment) to determine the levels of cytokines (IL-1 β, TNF, IFN-γ and IL-10), and they were carried out in the supernatant of the culture of mononuclear cells stimulated with antigen soluble Leishmania at a concentration of 5 μg/mL.

Patients were observed for follow-up every 2 weeks during the first month, every month until day 90, an assessment at 120 days, and 6 months after therapy (D180). Patients who did not return for follow-up were asked to return or were visited at home within 7 days of the missed appointment.

Recorded clinical parameters included the location of the largest lesion, the number of lesions, the size of the largest lesion and the presence of regional lymphadenopathy. 

Ulcers were measured with a standardized caliper and photographed at the initial visit and at each follow-up visit. Clinical and laboratory adverse events (AEs) were classified according to the Common Terminology Criteria for Adverse Events [21].

### 2.7. Clinical Endpoint Criteria

The primary endpoint (final cure) was 180 days after starting therapy (D180). The secondary endpoint (initial cure) was 90 days after starting therapy (D90). Time to cure in days and clinical and laboratory AEs were recorded. Healing was defined by complete re-epithelialization without raised edges, infiltrations or crusting of all lesions. Failure was defined as the presence of an active ulcer or healed lesion but with raised edges. All patients who failed on D90 received MA at a dose of 20 mg/kg/day for 30 days or amphotericin B (total dose 0.5 to 1 mg/kg) as recommended by the Ministry of Health of Brazil.

### 2.8. Statistical Analyses

All statistical analyses were performed using the Statistical Package for the Social Sciences software, version 20.0 (IBM Inc., Chicago, IL, USA). Numerical variables were analyzed descriptively and presented as the mean, standard deviation and [5] median, categorical variables were analyzed using the chi-square test, continuous variables were analyzed using the Student’s *t*-test, variables that were not normally distributed were analyzed by non-parametric tests and laboratory and immunological tests were analyzed by the Wilcoxon test. *p* < 0.05 was considered statistically significant. 

### 2.9. Ethics

Prior to study enrollment, written, informed consent was obtained from all patients. This study was approved by the Ethics Committee of the Professor Edgard Santos University Hospital—Federal University of Bahia (number 3,377,911).

## 3. Results

A total of 32 patients with CL from January 2020 to January 2022 were included in this study. Demographic and clinical characteristics are shown in Table 1. There was no loss to follow-up. There was no difference between the two groups. But males predominated in the G-CSF group (59%), while females predominated in the control group (53.3%). The main lesion was considered the one with the largest diameter; it was localized in the lower limbs in 76.5% and 60% of G-CSF and control group, respectively. All patients had a confirmed diagnosis of CL.

### 3.1. Efficacy

The cure rate at day 90 was 53% in the G-CSF group, while in the placebo group it was 47% (Table 2). At the final assessment (day 180) it increased to 65% in the G-CSF group, while in the placebo group it was 47%, but the difference was not significant. Healing time was lower in the G-CSF, but this was not significant. Relapses were uncommon, affecting only one subject in each group. Figure 1 and Figure 2 shows the therapeutic outcome of patients treated with MA + G-CSF and MA + placebo respectively. .

### 3.2. Cytokine Levels

The levels of IFN-γ, TNF-α, IL-1β and IL-10 before and during therapy in the two groups of patients are shown in Figure 3. No difference was found between the groups. TNF-α showed a decrease on day 15 during treatment in the G-CSF group (Figure 3B). We also observed a decrease in IL-10 in the G-CSF group (Figure 3D). 

### 3.3. Safety

Intradermal G-CSF was well tolerated. Four patients (23%) in the G-CSF group and three subjects (20%) in the placebo group complained of local and weak pain within 48 h. We did not observe any hematological or biochemical abnormalities in the group treated with G-CSF. Mild and transient systemic side effects such as arthralgia and/or myalgia (20%), headache (13%) and fever (7%) were observed in the placebo group, but these side effects also occurred less frequently in the G-CSF group. Nausea was found only in the placebo group. No patients needed to interrupt treatment.

## 4. Discussion

The increasing resistance of the parasite to antileishmanial drugs suggests that the currently used monotherapy needs to be revised. The rationale behind combination therapy is a faster cure, shorter duration of therapy and lower dose requirement, reducing costs and preventing the emergence of drug resistance [22]. To date, there are no published studies on the use of the MA and G-CSF combination in the treatment of CL. Studies performed in the same area with an immunomodulator (GM-CSF) showed its efficacy in patients with CL, both subcutaneously and topically [11,12].

G-CSF proved to be beneficial in toxic epidermal necrosis in subjects with or without neutropenia [23], as well in children with dystrophic epidermolysis bullosa [16], probably by attenuating CD8^+^ cytotoxicity as well by accelerating the healing process. The use of G-CSF in the experimental infection (bacterial and fungal infections) of non-neutropenic animals showed significant benefits after administration alone or in combination with antibiotics [24]. However, there are no data in the literature regarding the use of G-CSF in experimental or human CL. In our study, we did not show a significant increase in the cure rate with the use of G-CSF on D90 (53%) or D180 (65%). It is known that CL caused by *L. braziliensis* has a higher rate of therapeutic failure compared to CL caused by *L. panamensis* or *L. guyanensis* [5]. In fact, in the last decades, clinical trials published in the endemic area of Corte de Pedra have shown a cure rate ranging from 45% to 53% in patients with CL treated with MA [11,25]. 

We did not find statistical differences, either inter-group or between groups, in the cytokines that we analyzed. This somehow suggests that the application of intralesional G-CSF does not have systemic effects or the dose that was used had no systemic effect. The systemic use of G-CSF inhibits the production of IL-1β, IL-12, interferon (IFN)-γ and tumor necrosis factor (TNF)-α but increases serum levels of IFN-α and IL-10 [17]. In our study, we observed a decrease in TNF production by PBMC in the G-CSF group that could be beneficial at the tissue level to decrease inflammation favoring epithelization. However, we were not able to evaluate local cytokine production in our patients. The literature has shown that the use of G-CSF is well tolerated, although it can present mild to severe side effects such as anaphylaxis, and most adverse effects are related to systemic use [26,27]. Our patients had mild adverse events, probably related to the use of MA, and did not require treatment interruption. None of the individuals treated with intralesional G-CSF in the study presented hematological or biochemical changes. These data indicate that the intradermal use of G-CSF is safe and favors its use in higher dosages in future trials, if necessary.

## 5. Conclusions

Despite the higher cure rate and shorter healing time observed in patients treated with MA and G-CSF, these differences did not achieve statistical significance. However, our study gives support for new trials using higher doses or different routes of application of G-CSF in association with the standard treatment of CL caused by *L. braziliensis* to raise the low cure rates.

## Figures and Tables

**Figure 1 pathogens-13-00301-f001:**
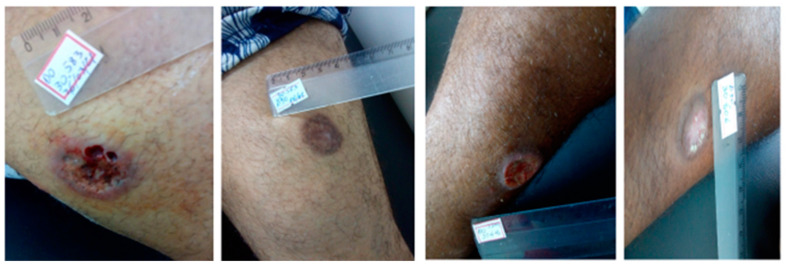
Clinical evolution of two patients treated with MA + G-CSF who progressed to cure on day 90 after starting treatment. Presentation of two patients from the G-CSF group D0 and D90 who were cured on day 90.

**Figure 2 pathogens-13-00301-f002:**
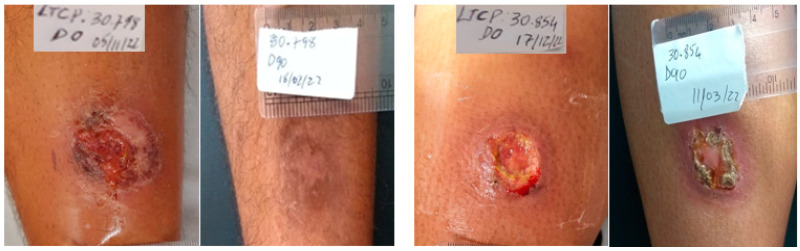
Clinical evolution of two patients treated with MA+placebo, one progressed to cure and the other failed and was treated on day 90 with MA for 30 days. Presentation of two patients from the placebo group, D0 and D90. One patient was cured on day 90, and the other failed.

**Figure 3 pathogens-13-00301-f003:**
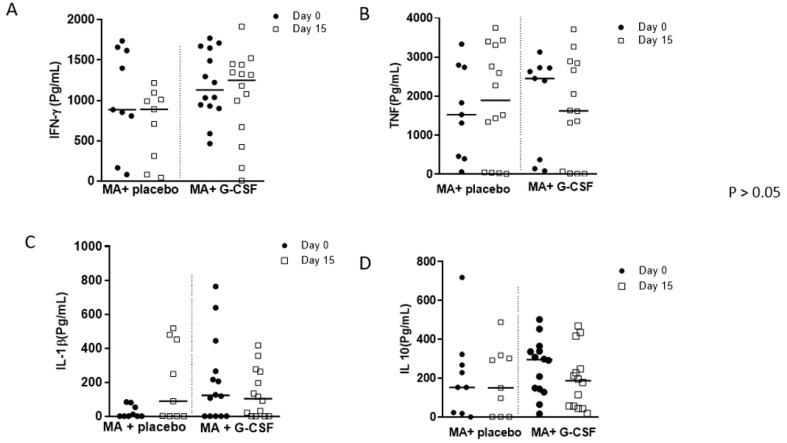
Cytokine levels in CL patients during MA treatment associated with intralesional G-CSF. MA: Meglumine Antimoniate, G-CSF: granulocyte colony-stimulating factor. The graphs (**A**–**D**) show the production of cytokines in patients with CL treated with MA associated with G-CSF (N = 14) or MA associated with placebo (N = 09). (**A**) IFN-γ, (**B**) TNF-α, (**C**) IL-1β, (**D**) IL-10. Cytokine levels were determined by ELISA in SLA-stimulated culture supernatants on days 0 and 15 during treatment. Wilcoxon test and Mann–Whitney test. (*p* < 0.05). No differences were found between groups.

**Table 1 pathogens-13-00301-t001:** Demographic, clinical and laboratory aspects of CL patients treated with MA + G-CSF or MA + placebo.

Variables	MA with G-CSF (Group A) n = 17	MA with Placebo (Group B) n = 15	*p*-Value
Gender: M (%)	10 (59%)	07 (46.6%)	0.37 *
Age (years, mean + SD)	31.6 ± 10.8	32.9 ± 12.8	0.94 **
Duration of illness 30–60 days >60–90 days	16 (94%) 1 (6%)	14 (93.3%) 1 (6.7%)	0.72 *
Location of the biggest lesion Cephalic segment Trunk Upper extremities Lower extremities	0 (0%) 0 (0%) 4 (23.5%) 13 (76.5%)	0 (0%) 2 (13.3%) 2 (13.3%) 11 (73.3%)	0.25 *
Number of lesions Single lesion Two lesions Three lesions	13 (76.5%) 3 (17.6%) 1 (6%)	9 (60%) 5 (33.3%) 1 (7%)	0.57 *
Largest diameter (mm²)	27.18 ± 6.7	27.47 ± 7	0.90 ***
Lymphadenopathy (%)	12 (70.6%)	9 (60%)	0.39 *
Positive PCR	16 (94%)	15 (100%)	0.53 *
Positive LST (%)	16 (94%)	15 (100%)	0.53 *

MA: meglumine antimoniate; M: median; SD: standard deviation. M: male; * Fisher Exact test; ** Student’s *t*-test; *** Mann–Whitney test; PCR: polymerase chain reaction; LST: Leishmania skin test.

**Table 2 pathogens-13-00301-t002:** Therapeutic outcome of CL patients treated with MA and G-CSF or MA and placebo.

Therapeutic Result	MA with G-CSF (Group A) n = 17	MA with Placebo (Group B) n = 15	*p*-Value
Cure on D90 (%)	9 (53%)	7 (47%)	0.50 *
Final cure rate (D180) (%)	11 (65%)	7 (47%)	0.40 *
Rescue therapy (%)	6 (35%)	7 (47%)	0.5 *
Healing time (days) range (M ± SD)	90 (57.5–1 × 35)	150 (50–2 × 10)	0.77 **
Relapse (%)	1 (6%)	1 (7%)	-
Irregular use (%)	0	0	-

M, median; SD, standard deviation; * Fisher Exact test; ** Mann—Whitney test.

## Data Availability

Publicly available datasets were analyzed in this study.

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
