# Peer review of "Pentavalent Antimony Associated with G-CSF in the Treatment of Cutaneous Leishmaniasis Caused by *Leishmania (Viannia) braziliensis"

_pathogens, 2024, doi:10.3390/pathogens13040301_

Round 1

Reviewer 1 Report

Comments and Suggestions for Authors

The short communication "Pentavalent Antimony Associated with G-CSF in the Treatment of Cutaneous Leishmaniasis caused by Leishmania (Viannia) braziliensis" provides a perspective of the use of meglumine antimoniate in association with G-CSF in the treatment of cutaneous leishmaniasis.

Although the results did not prove a higher efficiency of this treatment compared to the use of meglumine in monotherapy, it is interesting in the perspective that it sheds light on a new possible treatment that deserves further investigation.

I believe the authors should continue this research, increasing the number os participants.

Comments on the Quality of English Language

The English language needs minor corrections. In the general, the paper is quite well written.

Author Response

Thank you dear reviewer,

The comment is very pertinent to improve the quality of the manuscript. In perspective, we are already planning to increase the number of patients and the doses of G-CSF to see if we can have a satisfactory result, but at the same time we are facing several difficulties especially as the number of cases has been falling in the endemic area. We are working to improve the English level of the manuscript, thank you again.

Sincerely

Carvel

Reviewer 2 Report

Comments and Suggestions for Authors

This is a straight forward study in which the authors sought to obtain evidence of the benefit of including G-CSF to the treatment of cutaneous leishmaniasis. Using the dosage scheme that they outlined, the beneficial effect of G-CSF was negligible.

The study can be enhanced by included representative images of lesions with and without cure.

Author Response

 Dear reviewer, thank you for the comments that will contribute to improve the quality of the manuscript. We have included images of lesions cured and not cured at D90.  

Reviewer 3 Report

Comments and Suggestions for Authors

The current manuscript (Pentavalent Antimony Associated

with G-CSF in the Treatment of Cutaneous Leishmaniasis caused by Leishmania

(Viannia) braziliensis) by Suprien et al.

Generally it is simple study and easily designed. Some points need to clarify from the

authors:

1.      Table 1, contains detailed data about the patients in the experiment so, it needs to add it to the section materials and methods

2.      The route of treatement (intralesional), why? And why the authors didn't use G-CSF systemically?

3.      Figure 1 is unclear and not good, please supply and needs to add description of it

4.      Limitation of the study in using G-CSF needs more details with refrences and the causes of its use at the lesions

5.      Conclusion that G-CSF safety is not confirmed and the authors said no hematological changes was recorded but this was not in the materials or the results.

6.      The follow up of patients for 180 day, how? 

Author Response

Answer: Dear reviewer, thank you for the comments that will contribute to improve the quality of the manuscript.

  1. We reviewed the table. We also added more information at materials and methods section and improved its organization.
  2. 2. We used intralesional G-CSF because the literature shows that this route is safer and adverse effects are uncommon, as discussed in the text. Additionally, our group had good efficacy data in CL using another cytokine (GM-CSF) by intralesional route (references 11 and 12 of the manuscript).
  3. Figure 1 is now figure 3 and we have modified it, also including a description.
  4. In fact, it is to our knowledge the first time that G-CSF is used in CL. There are very few data about G-CSF use in others skin diseases, as already cited in the text. Please see lines 56-68.
  5. The safety of G-CSF in our study was assessed and described. Please see Materials and Methods – Study procedures. We also included further information in Results – item 3.3 Safety.
  6. Clinical trials in CL use 2 end points to access cure: 90 days after beginning the treatment (initial cure) and 180 days (final cure). We have explained this at materials and methods section.

Reviewer 4 Report

Comments and Suggestions for Authors

Dear authors,

the manuscript „Pentavalent Antimony Associated with G-CSF in the Treatment of Cutaneous Leishmaniasis caused by Leishmania (Viannia) braziliensis.”. Cutaneous leishmaniasis is important disease and it the examinations concerning its treatment should be made. Authors should take into account in the discussion section that not statistically significant result can be due to wrong time intervals. The cytokines have different  and often short half-lives.

Minor comments:

Line 65: Change “stimulating” into “activating”.

Line 72 and elsewhere: CD 8 + write + with upper index

Lines 75-78: Rewrite the sentence because it is difficult to understand.

Line 105: Student’s t-test.

Line 133 and elsewhere: Change “levels” into “concentrations”.

Line 167: Did authors means that it totally inhibits or only decrease?

Author Response

Answer: Dear reviewer, thank you for the comments that will contribute to improve the quality of the manuscript. We agree that cytokines vary in half-lives; however, it is not possible to verify it in CL subjects; we would have to take samples in several intervals from subjects that live in a rural area, raising the costs and ethical concerns. We decided to use the time intervals that were already used in the literature.

Line 65: We have changed (line 60)

Line 72 and elsewhere: We have corrected it.

Lines 75-78: We have modified the sentence

Line 105: We have corrected it (line 134)

Line 133 and elsewhere: We would like to keep “levels”. It is more widely used and, in our case, it is more appropriate than “concentrations”. In fact, we are measuring cytokines production in supernatants of peripheral blood mononuclear cells stimulated by Leishmania antigen. We are not measuring tissue concentration.

Line 167: Only decreases, we have corrected it, please see line 220.

Round 2

Reviewer 3 Report

Comments and Suggestions for Authors

The authors replied to all comments. The manuscript is accepted in the current form